# Physicians’ Views and Agreement about Patient- and Context-Related Factors Influencing ICU Admission Decisions: A Prospective Study

**DOI:** 10.3390/jcm10143068

**Published:** 2021-07-11

**Authors:** Stéphane Cullati, Thomas V. Perneger, Fabienne Scherer, Mathieu Nendaz, Monica Escher

**Affiliations:** 1Division of Palliative Medicine, Department of Readaptation and Geriatrics, University Hospitals of Geneva, Rue Gabrielle-Perret-Gentil 4, 1211 Geneva, Switzerland; stephane.cullati@unifr.ch (S.C.); fascherer@hotmail.com (F.S.); 2Division of Quality of Care, Department of Readaptation and Geriatrics, University of Geneva, 1211 Geneva, Switzerland; 3Population Health Laboratory, Department of Community Health, Faculty of Science and Medicine, University of Fribourg, 1700 Fribourg, Switzerland; 4Division of Clinical Epidemiology, University Hospitals of Geneva, 1211 Geneva, Switzerland; thomas.perneger@hcuge.ch; 5Unit of Development and Research in Medical Education, Faculty of Medicine, University of Geneva, 1211 Geneva, Switzerland; mathieu.nendaz@hcuge.ch; 6Division of General Internal Medicine, Department of Medicine, University Hospitals of Geneva, 1211 Geneva, Switzerland

**Keywords:** decision making, intensive care, admission, palliative care, triage, quality of care

## Abstract

Background: Single patient- and context-related factors have been associated with admission decisions to intensive care. How physicians weigh various factors and integrate them into the decision-making process is not well known. Objectives: First, to determine which patient- and context-related factors influence admission decisions according to physicians, and their agreement about these determinants; and second, to examine whether there are differences for patients with and without advanced disease. Method: This study was conducted in one tertiary hospital. Consecutive ICU consultations for medical inpatients were prospectively included. Involved physicians, i.e., internists and intensivists, rated the importance of 13 factors for each decision on a Likert scale (1 = negligible to 5 = predominant). We cross-tabulated these factors by presence or absence of advanced disease and examined the degree of agreement between internists and intensivists using the kappa statistic. Results: Of 201 evaluated patients, 105 (52.2%) had an advanced disease, and 140 (69.7%) were admitted to intensive care. The mean number of important factors per decision was 3.5 (SD 2.4) for intensivists and 4.4 (SD 2.1) for internists. Patient’s comorbidities, quality of life, preferences, and code status were most often mentioned. Inter-rater agreement was low for the whole population and after stratifying for patients with and without advanced disease. Kappa values ranged from 0.02 to 0.34 for all the patients, from −0.05 to 0.42 for patients with advanced disease, and from −0.08 to 0.32 for patients without advanced disease. The best agreement was found for family preferences. Conclusion: Poor agreement between physicians about patient- and context-related determinants of ICU admission suggests a lack of explicitness during the decision-making process. The potential consequences are increased variability and inequity regarding which patients are admitted. Timely advance care planning involving families could help physicians make the decision most concordant with patient preferences.

## 1. Introduction

Decisions to admit a patient to intensive care (ICU) are often complex [1,2]. The benefit of intensive care may be uncertain for patients with advanced disease [3] or for frail elderly patients [4]. Decisions are made in a time-pressured environment and require clinical judgment. Estimated survival probabilities are associated with physicians’ admission decisions [5], but patient- or context-related factors may also influence decisions [1,6,7,8].

ICU admission decisions for hospitalized patients involve clinical assessment of a critically ill patient and consideration of information provided by the ward physician. Although internists and intensivists share the same perception of each other’s roles [9], it is not known whether they consider the same factors in a similar manner when assessing a patient for intensive care. Lack of a shared mental model and poor communication are associated with deficient teamwork and higher levels of risk for patients [10,11,12]. Ideally, potential disagreements about the relevant determinants of an ICU admission decision would be identified and discussed between the physicians, and a decision would be reached through a collaborative process.

Research to date has enabled the identification of patient- and context-related determinants of ICU admission, but data are lacking about how physicians weigh these various factors and integrate them into the decision-making process [7,13]. Determining physicians’ views shortly after they assessed a critically ill patient for intensive care can provide a reliable evaluation of the actual determinants of an admission decision. This issue is particularly relevant for patients with advanced disease since patient- and context-related factors are more likely to influence admission to or refusal of intensive care. Although professional guidelines deem overtriage to be more acceptable than undertriage [2], a better understanding of physicians’ decision making is useful to inform the means of providing adequate levels of care to patients.

The first objective of this study was to determine which patient- and context-related factors internists and intensivists considered as influencing the decision to admit or not admit a medical inpatient to the ICU and to determine the degree of agreement between the two groups of physicians. The second objective was to determine if there were differences for patients with advanced disease compared to patients without advanced disease.

## 2. Methods and Materials

### 2.1. Design and Setting

This prospective study was conducted between August 2014 and August 2015 at the Geneva University Hospitals, Geneva, Switzerland, a tertiary care hospital including 156 internal medicine beds and 34 adult ICU beds. The Geneva Research Ethics Committee approved the study.

In this hospital, a critically ill inpatient is assessed by the internist, who decides whether intensive care may be appropriate and whether he will call the intensivist. After the internist has given all the relevant information to the intensivist on the ward, the latter personally evaluates the patient. The admission decision is made after the two physicians have discussed the case together.

### 2.2. Participants and Data Collection

We included all consecutive situations when an ICU consultation took place for a patient hospitalised in the Division of General Internal Medicine. Participants were the intensivist and the internist who personally evaluated the patient and made the admission decision. They provided written consent, and sociodemographic data were collected. Physicians were contacted within 12 h after the ICU consultation and answered a questionnaire administered by phone or by email. When necessary, a reminder was sent 3 days later.

Physicians were asked to rate the influence of patient- and context-related factors on the decision to admit or not admit the patient to intensive care. A list of factors was established based on a review of the literature and on a qualitative study we conducted [1] to determine the factors influencing ICU admission decisions. The instrument was pilot-tested with 2 internists and 2 intensivists who were not involved in the study. The question read as follows: “Several factors come into consideration when deciding whether to admit a patient to intensive care. Depending on the situation, they have a greater or a lesser weight. In the situation of this patient, what was the weight of the-listed below, if applicable?” The list included 13 factors: patient’s age, comorbidities, expected quality of life after the acute episode, patient’s preferences, family preferences, code status, physician’s knowledge of the patient, opinion of a specialist physician, opinion of a senior internist, workload on the medical ward, discomfort of intensive care for the patient, bed availability in the ICU, and time pressure. Factors were rated on a Likert scale ranging from 1 (negligible) to 5 (predominant). The option “not applicable” could be chosen for each factor.

Patient data were collected from the electronic medical file (see Appendix A Table A1). Advanced disease was defined as the presence of any of the following: metastatic cancer, active hematologic malignancy, chronic heart failure of NYHA stage III or IV and/or LVEF ≤ 20%, severe chronic obstructive pulmonary disease (FEV ≤ 50% or non-invasive ventilation or oxygenotherapy), severe chronic kidney disease (glomerular filtration rate < 30 mL/min), and liver cirrhosis Child–Pugh B or C.

### 2.3. Sample Size and Statistical Analysis

Modelling the ICU admission decisions was the main objective of the study, and it determined the estimation of the sample size. Assuming 8 potential predictors of the decision and an admission rate of 50%, we intended to include 160 patients. The observed admission rate was 70% and the sample size was increased to 200. Data were used for the analysis only if both physicians completed the questionnaire.

For each patient, we categorized the factors as having a major influence on the admission decision (score 4 or 5) or a non-significant one (scores 1–3, or not applicable). We cross-tabulated all factors, assessed by either physician, with the presence or absence of advanced disease, and compared the proportions by means of Fisher exact tests. We examined the degree of agreement using the kappa statistic on the dichotomized importance assessments, and followed Landis and Koch for interpreting kappa values [14]. Statistical significance was set at *p* < 0.05. We used SPSS version 24 software.

## 3. Results

Of 219 patients assessed for intensive care, 18 were excluded because of missing physician data, and 201 were included. Among the 201 included, 140 (69.7%) were admitted to the ICU. Patients had a median age of 67 years (inter-quartile range 56–77), and 128 (63.7%) were men. About half of the patients (*n* = 105; 52.2%) had an advanced disease, the most common being metastatic cancer and chronic obstructive pulmonary disease (Appendix A). The median number of days between hospital admission and ICU consultation was 3 (interquartile range 1–8). The main reasons for calling the ICU were respiratory failure (55.2%) and heart failure or shock (27.4%).

Thirty intensivists and ninety-seven internists participated in the study. A majority of intensivists were men (66%), and a majority of internists were women (61%). Intensivists were older than internists (mean age 38 and 30 years, respectively) and more experienced (mean years from graduation 12 and 7, respectively). The number of assessments by physicians ranged from 1 to 14 for intensivists and from 1 to 11 for internists. More than half the questionnaires (*n* = 213; 53%) were completed on the phone at the time of contact with the physicians. Overall, questionnaires were completed within 24 h for 127 (63.2%) intensivists and 143 (71.1%) internists, and within 72 h for 162 (80.6%) and 174 (86.6%), respectively.

### 3.1. Factors Rated as Important for the Admission Decision

Among 13 factors, physicians rated 0 to 11 factors per situation as having significantly influenced the decision to admit or not admit a patient to intensive care. On average, intensivists and internists considered that 3.5 (SD 2.4) and 4.4 (SD 2.1) factors, respectively (*p* ≤ 0.001), were important. Patient’s comorbidities, quality of life, preferences, and code status were most often mentioned (Table 1). Intensivists and internists infrequently rated the same factors as important for an admission decision. The determinants the two groups most often mentioned were patient’s comorbidities (*n* = 86; 42.8% of the decisions) and code status (*n* = 73; 36.3% of the decisions). Patient’s age, code status, and the opinion of a senior internist were rated as important more frequently by the internists, while the intensivists more often mentioned discomfort of intensive care for the patient. Inter-rater agreement on most factors was slight: kappa values ranged from 0.02 to 0.18 (Table 2). Fair agreement was found for family preferences (kappa 0.34) and patient’s comorbidities (kappa 0.21).

### 3.2. Factors Rated as Important in Patients with and without Advanced Disease

Most factors were rated as having a major influence on the admission decision in the same proportion of patients with and without advanced disease. Internists considered that comorbidities and discomfort of intensive care were more often influential for patients with advanced disease compared to patients without advanced disease (71.6% vs. 52.9%, *p* = 0.008; and 14.7% vs. 4.7%, *p* = 0.034, respectively). Intensivists more often rated workload on the medical ward as a predominant determinant in patients without advanced disease (25.9% vs. 12.1%, *p* = 0.015). Inter-rater agreement was low-slight for all but two factors (Table 2). There was moderate agreement about family preferences (0.42) for patients with advanced disease and fair agreement about patient’s comorbidities (0.32) for patients without advanced disease.

## 4. Discussion

The decision to admit or not admit a patient to intensive care can involve the consideration of patient- and context-related factors, in addition to strictly medical factors. This study is the first to our knowledge to evaluate intensivists’ and internists’ views on the determinants of an admission decision shortly after they assessed a critically ill patient for intensive care. This study shows that physicians rate many such factors as significantly influencing an admission decision. Patient-related factors, i.e., comorbidities, quality of life, and preferences, were rated as important more often than context-related factors. Intensivists and internists both estimated that code status had a major influence in about a third of the decisions.

Patient- and context-related determinants were mostly the same for patients with and without advanced disease. This finding is somewhat surprising. As intensive care may not be beneficial for patients with advanced disease, we would expect quality of life to influence the admission decision more frequently in this population than in patients without advanced disease. Similarly, although intensive care is more likely to represent a preference-sensitive decision for patients with advanced disease, no difference was found in the proportion of patients with and without advanced disease for whom the patient’s preferences were considered a major determinant of the decision. One reason for this finding may be that the patient’s values and preferences are not known at the moment of triage. A paucity of advance care planning has been pointed out [15,16], even though timely goals of care discussions are considered useful in decreasing ICU admission at the end of life [17,18,19]. Discussing patient preferences about levels of care, including care in the ICU, is a key part of advance care planning. It can provide physicians with valuable information and contribute to appropriate admission decisions.

Intensivists and internists assessed critically ill patients together and collaborated to determine whether intensive care was warranted. However, agreement beyond chance was low concerning the factors significantly influencing the admission decision in general, and after stratifying for patients with and without advanced disease. The best agreement concerned the influence of family preferences on the decision, especially in patients with advanced disease. Although moderate, it was higher than agreement about the impact of patient preferences.

Our data suggest a lack of communication during triage, whereas recommendations are that admission decisions should be made explicitly [2]. Variability between the intensivists and the internists in the interpretation of clinical situations could relate to differences in medical specialties, work environments, and cultures [8,20,21,22]. It could also be due to inter-individual differences in the appreciation of the various aspects of clinical situations, notwithstanding the medical specialty [23,24]. Similarly, low agreements were found among intensivists in an experimental study about the choice of life-sustaining treatments [25], and in two vignette-based studies about the benefit of ICU admission compared to care on the general ward [26] and about ICU admission for cancer patients [27]. The importance given to different determinants of admission decisions seems to vary among intensivists. Decisional patterns were identified with some intensivists giving high priority to patient age, while others privileged family views [28]. Lack of explicitness as observed in our study could increase variability and potential inequity regarding which patients are admitted to the ICU. There is currently no validated score to our knowledge for triage and no decision supporting framework, and physicians mainly rely on their clinical judgment when making admission decisions [19]. The development of decision-aid tools may help limit practice variations.

Among patient- and context-related factors, family preferences have been identified as significantly influencing admission decisions [1,24,28,29]. Pressure from family members belonged to the most common reasons given by intensivists for deviating from triage guidelines [27]. In our study, the best consensus was reached for family preferences, in particular for patients with advanced disease. Reasons for these findings can be manifold. Decisions are often complex for patients with advanced disease, and the opinion of the family is more likely to be expressly sought out by the physicians. It can be that discussions about goals of care are more explicit with families, or that relatives clearly express their preferences about the intensity of care for their loved one [1,28]. Of note, a time-limited admission to the ICU is an accepted strategy if the patient or family needs time to adapt [30]. Families can put so much pressure on physicians that patients are admitted to the ICU out of fear of litigation [1,27]. This raises the question of how appropriately the family plays the role of surrogate decision maker. Data on the topic are conflicting and suggest that families can be valuable health care surrogates or make choices not necessarily concordant with the patient’s values and preferences [1,31]. Interventions aimed at improving communication between patients and their families have the potential to support surrogate decision making [32,33,34].

This study has limitations. The physicians’ account of the influence of factors on the admission decisions was given retrospectively. However, it is hardly feasible to collect data at the moment of triage. We tried to minimize the risk of memory bias by contacting physicians twice a day, in the morning and in the evening, which means 12 h after the admission decision at the latest, and by offering them the option to complete the questionnaire on the phone, so the delays were kept as short as possible. We could test only a limited number of patient- and context-related factors. Other factors can influence decisions according to circumstances. However, all factors were considered important determinants by physicians for some admission decisions, which shows that they were clinically relevant. We included ICU consultations for patients hospitalised on general internal medicine wards. This was a single-centre study. The influence of context-related factors such as bed availability or workload on the ward may differ in other hospitals. Agreements between the intensivists and the referring physicians might be different in other clinical settings (e.g., in the emergency department). Since each physician may have assessed several patients, the observations are not independent. However, this concerned a few physicians only, and it is unlikely to affect the results. We cannot exclude that social desirability sometimes influenced physicians’ answers, but it is unlikely that the bias would be systematic or that it impacted the results significantly. If that were the case, we would expect a higher agreement between the physicians. Nonetheless, our findings bring novel information about physicians’ collaborative decision making for ICU admissions since previous studies assessed inter-professional shared decision making [35,36] or physicians’ end-of-life decision making in the ICU [37].

## 5. Conclusions

Intensivists and internists do not agree on the patient- and context-related factors influencing an ICU admission decision. Lack of transparency in the decision-making process can increase variability in admission decisions and lead to treatment inequity. Patients with advanced disease are particularly at risk as the benefit of intensive care can be difficult to determine for them. Effective advance care planning interventions by patients and their families could help physicians make the decision most concordant with patient values and preferences. Further studies are needed to evaluate the influence of poor agreement between physicians on practice variations during triage.

## Figures and Tables

**Table 1 jcm-10-03068-t001:** Opinions of the intensivist and the internist about the importance of specific factors in the decision to admit or not admit a patient to intensive care, in 201 decisions.

Factor	Rated as Important * by		*p* Value
	Intensivists	Internists	
	Proportion of Decisions (%)		
Patient’s age	19.4	40.8	<0.001
Patient’s comorbidities	59.2	63.7	0.36
Patient’s quality of life	48.3	53.2	0.36
Patient’s preferences	40.8	47.8	0.16
Family preferences	12.9	18.4	0.09
Code status	45.3	73.6	<0.001
Knowing the patient	24.9	27.4	0.63
Opinion of a specialist	18.4	24.4	0.14
Opinion of a senior internist	10.4	42.8	<0.001
Workload on medical ward	17.9	12.9	0.17
Discomfort of intensive care for the patient	24.9	10.4	<0.001
Availability of beds in the ICU	16.4	17.4	0.89
Time pressure	7.0	3.5	0.17

* 4 or 5 on 5-point scale from 1 (negligible) to 5 (predominant). Missing and non-applicable ratings were included and treated as less than 4 or 5.

**Table 2 jcm-10-03068-t002:** Agreement beyond chance between intensivist and internist about the influence of specific factors on the admission decision (dichotomized assessments), in 201 patients, and for patients with and without advanced disease.

Factor	All Patients *n* = 201	Patients	
		No Advanced Disease *n* = 96	Advanced Disease *n* = 105
	Kappa Statistic	Kappa Statistic	Kappa Statistic
Patient’s age	0.11	0.11	0.12
Patient’s comorbidities	0.21	0.32	0.09
Patient’s quality of life	0.03	0.01	0.04
Patient’s preferences	0.12	0.07	0.14
Family preferences	0.34	0.19	0.42
Code status	0.11	0.08	0.14
Knowing the patient	0.11	0.11	0.11
Opinion of a specialist	0.18	0.29	0.10
Opinion of a senior internist	0.02	0.01	0.04
Workload on medical ward	0.16	0.14	0.19
Discomfort of intensive care for the patient	0.16	−0.08	0.24
Availability of beds in the ICU	0.12	−0.04	0.27
Time pressure	0.05	0.16	−0.05

## Data Availability

All data relevant to the study are included in the article or uploaded as appendix. No additional data are available.

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
