# Peer review of "Physicians’ Views and Agreement about Patient- and Context-Related Factors Influencing ICU Admission Decisions: A Prospective Study"

_jcm, 2021, doi:10.3390/jcm10143068_

Round 1
Reviewer 1 Report
I have no further comments for the authors.
Author Response
thank you for reviewier this paper.
Reviewer 2 Report
Cullati et al conducted an analysis of internist and intensivist opinions of patient and contextual factors influencing ICU admission decision-making after joint ICU consultations of patients admitted to Geneva University Hospitals. They sought to test agreement between internists and intensivists. They found that there was poor agreement between internists and intensivists.
This finding is not surprising but important for clinical care, nonetheless. Disagreement between internists and intensivists places borderline patients at risk for mis-triage. There is a clear need to bridge these gaps to ensure patients who might benefit from ICU care will receive it. Strengths of the study include a well-written manuscript and its conceptual grounding within real patients. My concerns are noted below, from major to minor.
- Underlying reasons for poor agreement.
- It is not obvious to me that communication gaps are the main source for poor agreement between intensivists and internists. In many ways, it depends on how the survey is framed to participants. If the survey asks, “What were the factors that influenced the actual ICU admission decision for this patient?”, then poor agreement may reflect communication gaps as it may highlight that internists and intensivists did not communicate their rationale to each other. However if the survey instead asks, “In your opinion, which of these factors was most important to your ICU admission decision-making for this patient?”, then the question is phrased in a more personal manner where internist-intensivist communication is less important than the internal thought process of the individual participant. In this latter scenario, the lack of standardized strategies to evaluate ICU benefit is likely to be a more influential driver than communication gaps. I think a better framing is used by the Authors in paragraph 2 of the Introduction when they discuss mental models.
- The Abstract’s last sentence states that advanced care planning might help. However, this idea does not align with the data collected from this study. While, yes, advanced care planning might be helpful overall, this study does not directly assess whether prior advanced care planning might be influential. In fact, internists and intensivists did not even strongly agree that family preferences were influential. Thus, there is no evidence from this study that suggests more advanced care planning would impact agreement between internists and intensivists.
- Did the Authors ask participants whether they believed the patient should receive ICU care? It could be interesting to assess for disagreement between internists and intensivists to that question?
- The Authors highlight the mean number of factors considered between intensivists and internists, but it is not clear whether this is the total number of factors or total number of important factors. The Abstract states important factors while the Results omits the word “important”. I think the strength or importance of the factors is an important consideration.
- Depending on the number of surveys performed per participant, is it possible to assess whether there is within-clinician reliability in factors? For instance, are certain factors more important to some clinicians compared to others?
- Methods/Results clarifications.
- Is inter-rated reliability measured using the 5-point Likert scale variable or the dichotomous important/not important variable
- Since this is a single center study, it would be helpful to provide the reader with some context as to how ICU admission decisions are made at the hospital. How formal is the ICU consultation process? Do intensivists and internists evaluate patients in person and speak directly?
- The Authors should note as a limitation that this is a single center study, since factors related to ICU admission, particularly the influence of contextual factors such as bed availability, may differ by hospital characteristics.
- How many surveys were conducted by phone vs email? What was the average time from consult to survey? What was the distribution of surveys per participant?
- Advanced disease. Some clinicians may treat advanced disease differently based on the underlying disease. For instance, patients with cancer are less likely to receive ICU care while data do not suggest this for other conditions, such as COPD or heart failure, which are, like metastatic cancer, irreversible, but still thought of differently. Was there more agreement between clinicians among patients with metastatic cancer?
- Abstract clarity. The Abstract Methods do not specify that this is a single-center study, that consecutive patients were analyzed, what types of consultants were involved, or that inter-rated reliability was being evaluated.
Author Response
Reviewer 2
Cullati et al conducted an analysis of internist and intensivist opinions of patient and contextual factors influencing ICU admission decision-making after joint ICU consultations of patients admitted to Geneva University Hospitals. They sought to test agreement between internists and intensivists. They found that there was poor agreement between internists and intensivists.
This finding is not surprising but important for clinical care, nonetheless. Disagreement between internists and intensivists places borderline patients at risk for mis-triage. There is a clear need to bridge these gaps to ensure patients who might benefit from ICU care will receive it. Strengths of the study include a well-written manuscript and its conceptual grounding within real patients. My concerns are noted below, from major to minor.
- Underlying reasons for poor agreement.
- It is not obvious to me that communication gaps are the main source for poor agreement between intensivists and internists. In many ways, it depends on how the survey is framed to participants. If the survey asks, “What were the factors that influenced the actual ICU admission decision for this patient?”, then poor agreement may reflect communication gaps as it may highlight that internists and intensivists did not communicate their rationale to each other. However if the survey instead asks, “In your opinion, which of these factors was most important to your ICU admission decision-making for this patient?”, then the question is phrased in a more personal manner where internist-intensivist communication is less important than the internal thought process of the individual participant. In this latter scenario, the lack of standardized strategies to evaluate ICU benefit is likely to be a more influential driver than communication gaps. I think a better framing is used by the Authors in paragraph 2 of the Introduction when they discuss mental models.
Response: we agree with the reviewer and we thank him for his thoughtful comments. The introductory comment and phrasing of the question were neutral and descriptive (see below). However, lack of a shared mental model and communication gaps can both explain our findings, and they are not mutually exclusive. Our findings point to the risk of decisional variability associated with a lack of explicitness. We modified the conclusion of the abstract in order to convey this main message only.
Abstract – Conclusion:
Poor agreement between physicians about patient- and context-related determinants of ICU admission suggests a lack of explicitness during the decision making process. Potential consequences are increased variability and inequity regarding which patients are admitted.
Questionnaire:
Several factors can be considered to make the decision to admit or not a patient to intensive care. Depending on the situation, they have a greater or a lesser weight. In the situation of this patient, what was the weight of the - listed below (following factors), if applicable?
In French : « Plusieurs facteurs entrent en considération pour prendre la décision d’admettre ou pas un patient aux soins intensifs. Selon les situations, ils ont un poids plus ou moins importants. Dans la situation de ce patient, quel a été le poids des facteurs listés ci-dessous, si applicable ? »
- The Abstract’s last sentence states that advanced care planning might help. However, this idea does not align with the data collected from this study. While, yes, advanced care planning might be helpful overall, this study does not directly assess whether prior advanced care planning might be influential. In fact, internists and intensivists did not even strongly agree that family preferences were influential. Thus, there is no evidence from this study that suggests more advanced care planning would impact agreement between internists and intensivists.
Response: It is true that our study does not assess whether advance care planning can impact agreement between physicians, and this was not an objective. The last sentence of the Abstract is phrased as a hypothesis and describes an option worth exploring, not so much to increase agreement in general, but to guide the decision making and decrease potential variability. We were specific about 1) “Timely advance care planning”: discussing patient’s preferences well in advance can provide physicians with meaningful information about the appropriateness of intensive care; and about 2) the importance of involving families in advance care planning, so they can act as valuable healthcare surrogates. Timeliness and trustworthy surrogates are especially important for patients with advanced disease for whom admission to intensive care is more likely to represent a preference-sensitive decision. Our findings show that these patients’ preferences were not considered more often compared to those of patients without advance disease. In our study agreement about family preferences, even though moderate, was stronger for patients with advanced disease. This finding adds to the current literature about family’s influence on admission decisions which is discussed in the discussion section.
- Did the Authors ask participants whether they believed the patient should receive ICU care? It could be interesting to assess for disagreement between internists and intensivists to that question?
Response: We asked participants whether they were satisfied with the way the decision was reached, and with the admission decision itself. Intensivists and internists were generally satisfied with both the process and the decision. Internists were more satisfied than intensivists. However, we think this question is out of the scope of the study. The objective was to focus on patient- and context-related factors and to describe which factors physicians considered as significantly influencing the decision and assess the degree of agreement on these factors.
- The Authors highlight the mean number of factors considered between intensivists and internists, but it is not clear whether this is the total number of factors or total number of important factors. The Abstract states important factors while the Results omits the word “important”. I think the strength or importance of the factors is an important consideration.
Response: this is the total number of important factors. The title of this sub-section is “Factors rated as important for the admission decision” (see Results section), and the first sentence states “Among 13 factors, physicians rated 0 to 11 factors per situation as having significantly influenced the decision”. Still, we completed the sentence describing the mean number of factors according to the intensivists and to the internists.
“On average intensivists and internists considered that 3.5 (SD 2.4) and 4.4 (SD 2.1) factors, respectively (p = <0.001) were important.”
- Depending on the number of surveys performed per participant, is it possible to assess whether there is within-clinician reliability in factors? For instance, are certain factors more important to some clinicians compared to others?
Response: This is an interesting question. However, within-clinician reliability cannot be assessed as each physician saw any given patient only once, and each physician saw a different set of patients.
- Methods/Results clarifications.
- Is inter-rated reliability measured using the 5-point Likert scale variable or the dichotomous important/not important variable
Response: Thank you for this comment. The reported kappa statistics were done on the dichotomized assessments. We clarified the Methods and Table 2 header.
“We examined the degree of agreement using the kappa statistic on the dichotomized importance assessments, and followed Landis and Koch for interpreting kappa values”
Table 2. . Agreement beyond chance between intensivist and internist about the influence of specific factors on the admission decision (dichotomized assessments), in 201 patients, and…
- Since this is a single center study, it would be helpful to provide the reader with some context as to how ICU admission decisions are made at the hospital. How formal is the ICU consultation process? Do intensivists and internists evaluate patients in person and speak directly?
Response: We described the consultation process in our hospital in more details in the Methods (sub-section “Design and setting”).
“In this hospital, a critically ill inpatient is assessed by the internist, who decides whether intensive care may be appropriate and whether he will call the intensivist. After the internist has given all the relevant information to the intensivist on the ward, the latter personally evaluates the patient. The admission decision is made after the two physicians have discussed the case together.”
- The Authors should note as a limitation that this is a single center study, since factors related to ICU admission, particularly the influence of contextual factors such as bed availability, may differ by hospital characteristics.
Response: We added this limitation in the Discussion.
“This was a single-centre study. The influence of context-related factors such as bed availability or workload on the ward may differ in other hospitals.”
- How many surveys were conducted by phone vs email? What was the average time from consult to survey? What was the distribution of surveys per participant?
Response: Detailed data about the completion of the questionnaires are the following:
Completion occurred:
- at the time of contact with the physician for 87 (43.3%) of the intensivists and 126 (62.7%) of the internists; hence 213 (53%) questionnaires were completed by phone.
- within 24 hours for 127 (63.2%) and 143 (71.1%), respectively;
- within 48 hours for 143 (71.1%) and 153 (76.1%), respectively;
- within 72 hours for 162 (80.6%) and 174 (86.6%), respectively.
Distribution of surveys:
- among 30 intensivists: 1 to 14 assessments per physician
- among 97 internists: 1 to 11 assessments per physician.
We added this information in the Results.
“Number of assessments by physician ranged from 1 to 14 for intensivists, and from 1 to 11 for internists. More than half the questionnaires (n = 213; 53%) were completed on the phone at the time of contact with the physicians. On the whole, questionnaires were completed within 24 hours for 127 (63.2%) intensivists and 143 (71.1%) internists, and within 72 hours for 162 (80.6%) and 174 (86.6%), respectively.”
- Advanced disease. Some clinicians may treat advanced disease differently based on the underlying disease. For instance, patients with cancer are less likely to receive ICU care while data do not suggest this for other conditions, such as COPD or heart failure, which are, like metastatic cancer, irreversible, but still thought of differently. Was there more agreement between clinicians among patients with metastatic cancer?
Response: there were only 22 patients with metastatic cancer as an underlying disease. The estimation in this small group was very imprecise, the confidence intervals on the kappas were very wide. We do not think it is useful to add a table with results that cannot be really interpreted. Furthermore, the fact that cancer patients may be less likely to be admitted doesn’t mean that the physicians should agree more or less strongly about non-medical factors, in our opinion. In this sample, admission to the ICU was only slightly less likely (13/22, 59%) in patients with metastatic cancer than in the other subgroups (other serious illness, 67/94, 71%, and no serious illness, 60/85, 71%, p=0.52 for the comparison of the 3 groups).
- Abstract clarity. The Abstract Methods do not specify that this is a single-center study, that consecutive patients were analyzed, what types of consultants were involved, or that inter-rated reliability was being evaluated.
Response: We added this information in the Abstract, section Method.
“This study was conducted in one tertiary hospital. Consecutive ICU consultations for medical inpatients were prospectively included. Involved physicians, i.e. internists and intensivists, rated the importance of 13 factors for each decision on a Likert scale…”
“We cross-tabulated these factors by presence or absence of advanced disease, and examined the degree of agreement between internists and intensivists using the kappa statistic.”

This manuscript is a resubmission of an earlier submission. The following is a list of the peer review reports and author responses from that submission.
Round 1
Reviewer 1 Report
The authors of this paper are interested in the factors that can influence the decision to admit or not to intensive care by comparing the points of view of intensivists and internists.
This is a single-center prospective study that took place in Geneva between 08.2014 and 08.2015. After consultation on the decision to admit or not the patient in intensive care, a questionnaire (13 questions) from previous qualitative research was sent to the doctors who had participated in this consultation within 12 hours.
Regarding the results that may influence the admission or not of a patient in intensive care, these are relatively standard compared to the data in the existing literature, in particular as regards the presence of co-morbidity, quality of life, preferences of patients and the pathology leading to the consultation.
The authors of the paper suggest that there is probably a lack of communication between physicians and probably a lack of anticipation of a potential ICU admission which could be offered as part of advance care planning.
The paper is ethically interesting but offers no solutions to problems that are well known.
Reviewer 2 Report
This paper addresses an important issue, which is medical practice variation around the matter of ICU admissions. As written, this reviewer has some outstanding questions that would help contextualize the study in existing literature and also help readers make sense of the results.
In the front end of the paper, there is minimal contextualization in terms of the large, interdisciplinary literature on variations in medical practice—how providers make different decisions about the same clinical case. Thus, it is unclear how this paper fills a gap. Based on existing knowledge, it is entirely unsurprising that intensivists and internists would make difference assessments of the same clinical case. Yet this is not clarified in the introduction. What is the gap being addressed by this study? Is it just that this study hasn’t been done before and is therefore needed (that is not a compelling justification). What are the risks of admitting too much or not enough to the ICU? What are the predictors of over- and under-admission? What are the hypotheses that drive the study design? In short, the study is set up as if it is far more novel than it is.
In terms of study design, these types of studies often use vignettes or experiments to control for variations in the clinical case. In this case, they are asking for retrospective account for the factors that influenced an ICU admission decision. There are limitations in this retrospective account design, specifically around social desirability and legal protections of decisions. These trade-offs, and how the current study design matters for how this study fits in with the larger research, are not addressed and this should be further contextualized.
In terms of analysis, are there concerns that there were only 30 intensivists compared to 97 internists? It would be appropriate to present data on the distribution of intensivists across cases so there can be an assessment of whether or not there should be analytic controls for certain physicians seeing many patients in the study. In other words, how to do we know there isn’t an effect of one physician seeing half the patients so that perhaps the result is less about intensivists but more about that specific physician? This is entirely unaddressed.
Table 2—please comment on how you contextualized these results, and what you consider significant in terms of the Kappa statistics.
Discussion and conclusion—again, it is not surprising that there are differences in these outcomes and the paper would benefit from more set up earlier in the manuscript in terms of hypotheses and what is missing in terms of extant literature.
Reviewer 3 Report
I find the general objective hard to grasp. The objective mentioned in the abstract is even more confusing and is lacking some clinical need.
The whole manuscript (abstract, introduction, results, discussion, conclusion) should be more straightforward in defining, analyzing and answering a real clinical need.
This could be the final goal to standardize ICU admission for instance. Are there any scores described in the literature? Please discuss.
What about the amount of free ICU beds? Was this analyzed as well?